# Evaluation of the Therapeutic Potential of Traditionally-Used Natural Plant Extracts to Inhibit Proliferation of a HeLa Cell Cancer Line and Replication of Human Respiratory Syncytial Virus (hRSV)

**DOI:** 10.3390/biology13090696

**Published:** 2024-09-05

**Authors:** Ellie N. Brill, Natalie G. Link, Morgan R. Jackson, Alea F. Alvi, Jacob N. Moehlenkamp, Morgan B. Beard, Adam R. Simons, Linden C. Carson, Ray Li, Breckin T. Judd, Max N. Brasseale, Emily P. Berkman, Riley K. Park, Sedna Cordova-Hernandez, Rebecca Y. Hoff, Caroline E. Yager, Meredith C. Modelski, Milica Nenadovich, Dhruvi Sisodia, Clayton J. Reames, Andreas G. Geranios, Sean T. Berthrong, Anne M. Wilson, Ashlee H. Tietje, Christopher C. Stobart

**Affiliations:** 1Department of Biological Sciences, Butler University, 4600 Sunset Ave., Indianapolis, IN 46208, USAsberthro@butler.edu (S.T.B.); atietje@butler.edu (A.H.T.); 2Department of Chemistry and Biochemistry, Butler University, 4600 Sunset Ave., Indianapolis, IN 46208, USA; amwilson@butler.edu; 3Interdisciplinary Program in Public Health, Butler University, 4600 Sunset Ave., Indianapolis, IN 46208, USA

**Keywords:** cancer, apoptosis, antiviral, phytochemistry, traditional medicine, respiratory syncytial virus

## Abstract

**Simple Summary:**

Natural plant products have been used medicinally for thousands of years by Native Americans in the United States to treat a wide array of ailments. However, there remains a need to investigate the therapeutic potential or effectiveness of these traditional approaches as little remains known. In this study, we evaluated the therapeutic potential of aqueous extracts prepared from four plants traditionally used in Indiana, USA, to inhibit cancer cell proliferation and infection with human respiratory syncytial virus (hRSV), a major respiratory pathogen of infants and the elderly.

**Abstract:**

Traditional approaches employing natural plant products to treat a wide array of ailments have been documented and described for thousands of years. However, there remains limited scientific study of the therapeutic potential or effectiveness of ethnobotanical applications. Increases in the incidence of cancer and emerging infectious diseases demonstrate a growing need for advances in the development of therapeutic options. In this study, we evaluate the therapeutic potential of aqueous extracts prepared from four plants, purple aster *(Symphyotrichum novae-angliae* (L.) *Nemsom*), common sage (*Salvia lyrata* (L.)), northern spicebush (*Lindera benzoin* (L.) *Blume*)*,* and lamb’s ear (*Stachys byzantina* (K.) *Koch*)) traditionally used in Native American medicine in Indiana, USA. Using a combination of cytotoxicity assays, immunofluorescence microscopy, and antiviral assays, we found that sage and spicebush extracts exhibit cytotoxic and antiproliferative effects on HeLa cell proliferation and that sage, spicebush, and aster extracts were capable of significantly inhibiting human respiratory syncytial virus (hRSV), a major respiratory pathogen of infants and the elderly. Chemical analysis of the four extracts identified four major compounds which were subsequently evaluated to identify the responsible constituents in the extracts. While none of the identified compounds were shown to induce significant impacts on HeLa cell proliferation, two of the compounds, (1S)-(-)-Borneol and 5-(hydroxymethyl)-furfural, identified in sage and spicebush, respectively, were shown to have antiviral activities. Our data suggest that several of the extracts tested exhibited either anti-proliferative or antiviral activity supporting future further analysis.

## 1. Introduction

Botanical products and preparations have been used for millennia to treat many different ailments worldwide. Plants and their chemical constituents have provided the foundation for a multitude of products that we have today including up to half of all current pharmaceuticals [1]. Approximately 80% of people worldwide continue to primarily rely on plant-based medicines [2]. However, it is believed that as few as 10% of known plants today have been investigated for their therapeutic potential despite the high prevalence of plant-based medicines [1,2]. Increasingly, there has been a push to investigate and explore the wealth of knowledge shared within indigenous cultures. In Indiana, a U.S. state in the Midwest, indigenous peoples such as the Miami have maintained a rich history and culture of using the products of the land for many applications including treating disease [3]. Despite advances in science and modern medicine, there remains a significant need for continued development of therapeutics to treat major causes of disease such as cancer and infectious diseases.

Cancer is the second leading cause of death in the United States (after cardiovascular events) and is a major cause of both morbidity and mortality worldwide [4,5]. Cancer is associated with uncontrollable cell growth and proliferation, may impact any tissue in the human body, and can be caused by an incredibly diverse number of possible genetic modifications, making both treatment and development of targeted therapeutics difficult [6,7]. Furthermore, additional challenges to treating cancer include overcoming inequities in access to healthcare, high costs of treatments, and the limited availability of specialized facilities with cancer expertise [8]. Current treatments for cancer remain largely invasive with significant side effects and often include a combination of both chemotherapy and radiation. With rising cases of cancer today, identifying novel compounds that can limit the growth and proliferation of cancer cells remains a major need [9].

Established and emerging infectious diseases are collectively responsible for approximately 20% of all deaths worldwide [10]. In 2019, a novel coronavirus (SARS-CoV-2) emerged in Wuhan, China and quickly spread, triggering a worldwide pandemic. To date (as of July 2024), the COVID-19 pandemic has resulted in approximately 775 million confirmed cases and over 7 million deaths [11]. Despite a wealth of knowledge about existing coronavirus biology, efforts to identify and develop safe and effective therapeutics for COVID-19 have proven challenging. Numerous studies turned to ethnobotany and traditional medicines as a source of potential therapeutics [12,13,14]. For less studied viruses, such as human respiratory syncytial virus (hRSV), which is a leading viral cause of infant mortality worldwide, often there remain limited therapeutic options and the default treatment remains palliative care.

Identifying plant phytocompounds with therapeutic potential to treat significant causes of disease such as viruses and cancer can expand treatment options and provide a platform for further development and optimization. In this study, we analyzed extracts prepared from four plants used traditionally in Indiana (*Symphyotrichum novae-angliae* (L.) *Nemsom, Salvia lyrata* (L.), *Lindera benzoin* (L.) *Blume*, and *Stachys byzantina* (K.) *Koch*) for anti-cancer proliferative properties against the HeLa cell cancer line and antiviral activity against hRSV. HeLa cells are a cervical cancer cell line that has been extensively studied and has been used in previous studies to evaluate the therapeutic potential of natural agents [15,16,17]. There remain no commercially available therapeutics specific to treating active hRSV cases. Our studies aim to identify significant biological activities towards both HeLa cells and hRSV that may have anti-cancer and anti-viral therapeutic potential for further investigation.

## 2. Materials and Methods

### 2.1. Cells and Viruses

HEp-2 cells (ATCC CCL-23), a cervical cancer cell line derived from HeLa contamination, were maintained and cultured at 37 °C under 5% CO_2_ in Dulbecco’s Minimal Essential Media (DMEM) supplemented with 10% fetal bovine serum (FBS) and 50 μg/mL penicillin, 50 μg/mL streptomycin, and 2.5 μg/mL amphotericin B (PSA). Antiviral studies were performed using an hRSV strain A2 which was modified to express a far-red fluorescent reporter protein, monomeric Katushka 2 (mKate2), that has been previously described [18].

### 2.2. Plant Extract Preparation

Leaves, flowers, and stems of purple aster (*Symphyotrichum novae-angliae* (L.) *Nemsom*), common sage (*Salvia lyrata* (L.)), northern spicebush (*Lindera benzoin* (L.) *Blume*), and lamb’s ear (*Stachys byzantina* (K.) *Koch*) were identified by an ethnobotanist and obtained at the Whitley County Historical Museum in Whitley County, Indiana, United States (41°9′33.157″ N, 85°29′26.107″ W). Plant specimens obtained from the plants examined in this study were properly preserved in the Friesner Herbarium at Butler University under the accession numbers 20240115 and 20240117–20240120 and are available for examination upon request. The plant products were dried using a mechanical convection oven at approximately 32 °C (90 °F) before the leaves and flowers were extracted and coarsely chopped. Aqueous extracts were prepared by adding sufficient water heated to 95 °C to prepare concentrated solutions (0.08 g/mL aster, 0.10 g/mL sage, 0.13 g/mL spicebush, and 0.07 g/mL lamb’s ear). The extracts were steeped in the heated water for approximately 10 min. The extracts were then initially gravity filtered using coffee filters, before being treated with 10 mJ of UV light and filter-sterilized (0.2 micron) prior to use in cell culture. Extracts were stored long-term at −80 °C until use.

### 2.3. GC-MS Analysis of Plant Extracts and Testing of Major Compounds

A 1 mL aliquot of each aqueous plant solution was spiked with 10 mL methanol and extracted with 500 mL ethyl acetate in the presence of 200 mg anhydrous NaCl. The organic phase removed and analyzed by gas chromatography–mass spectroscopy (GC-MS) following an adapted procedure previously described [19]. Prominent hits were identified using a library database and were purchased for direct testing at the relative concentrations identified through comparison to the loaded methanol standard. The compounds tested in this study were 1,8-cineole (Thermo Scientific Chemicals, CAS 470-82-6, Avocado Research Chemicals Ltd. (part of Thermo Fisher Scientific), Lancashire, UK), 5-(hydroxymethyl)-2-furaldehyde (Thermo Scientific Chemicals, CAS 67-47-0, Avocado Research Chemicals Ltd. (part of Thermo Fisher Scientific), Lancashire, UK)), D-(+)-melezitose hydrate (TCI America, CAS 207511-10-2, Portland, OR), and (1S)-(−)-Borneol (Thermo Scientific Chemicals, CAS 464-45-9, Avocado Research Chemicals Ltd. (part of Thermo Fisher Scientific), Lancashire, UK). Solutions were prepared of each in cell culture medium at a range of dilutions proportional to those identified through GC-MS in the plant extracts. The 1× concentrations of the chemicals found by GC-MS were 5.1 mM cineole, 4.8 mM furfural, 1 mM melezitose, and 7.7 mM borneol.

### 2.4. Cytotoxicity (MTS) Assay

HEp-2 cells were plated at a density of 20,000 cells per well and incubated for 24 h prior to treatment with different concentrations of plant extracts for an additional 24 h. After 24 h, the supernatant containing medium and dilutions of plant extracts was removed and new culture medium containing the MTS solution was added as directed (Promega CellTiter 96 Aqueous One Solution Cell Proliferation Assay). Absorbances were collected after 2 h incubation using a spectrophotometer at a wavelength of 490 nm. Cell viability was determined using the following formula: % viability = [(A_t_/A_s_) × 100%], where A_t_ is the absorbance of the concentrated extract and A_s_ is the concentration of the solvent.

### 2.5. Immunofluorescence (IF) Microscopy

Cells were plated on coverslips at a density of 0.1 × 10^6^ cells/well and incubated for 24 h prior to the addition of plant extracts at a 1:10 dilution in complete DMEM. After an additional 24 h, the coverslips were fixed, permeabilized, and treated with DAPI (4′,6-diamidino-2-phenylindole) at a concentration of 1 μg/mL, TRITC-conjugated Phalloidin (1:300 dilution), and a FITC-conjugated anti-tubulin antibody (1:200 dilution). Multiple images were obtained from random viewing areas within each coverslip using a Leica DM5500 fluorescence microscope. Images were scored blindly for the relative percentage of normal cells, those with altered nuclear morphology, and mitotic cells, as well as for cytoskeletal disruption for each treatment condition.

### 2.6. Caspase-3/-7 Activity

HEp-2 cells were plated at a density of approximately 20,000 cells per well and incubated for 24 h prior to treatment with different concentrations of plant extracts for an additional 4 h. After 4 h, the Caspase-Glo 3/7 Reagent was added as directed (Promega Caspase-Glo 3/7 Assay). Luminescence values were collected after 30 min incubation. Values reported are after subtracting the value from a blank reaction without cells and comparing to untreated cell controls.

### 2.7. Virus Inactivation Assay

HEp-2 cells were treated with mixtures of complete DMEM containing different dilutions of plant extracts combined with recombinant hRSV strain A2 expressing mKate2 at a multiplicity of infection (MOI) of 0.05 infectious particles per cell. The number of infected cells as indicated by red fluorescent foci using a Leica DMIL microscope with a Texas Red filter was quantified and compared to untreated infection controls as previously described [20].

### 2.8. Statistical Analysis

Cytotoxicity data were analyzed with analysis of covariance (ANCOVA) to test for the main effects of dosage and plant extract type as well as the interaction of the effects, which tested if the effect of different plant extracts depended on concentration. Data analysis was conducted using RStudio (ver. 3.6.0) [21]. ANCOVA was conducted using the CAR package [22]. When an overall interaction was found to be significant, contrasts were run to test if specific plant extracts differed overall from control (water). Slopes of plant extract effect by dosage were tested using the emtrends function from the emmeans package (ver. 1.10.04) [23]. Apoptosis, cytoskeleton structure, and mitosis scores were analyzed using ANOVA [22]. When the overall ANOVA model testing for differences in plant extract effects was significant, posthoc multiple comparisons (Tukey’s HSD) were conducted to test if specific plant extract effects differed from each other or water controls. For all analyses, data were examined if they met the assumptions of ANOVA (normality, equal variances). If the data did not meet those assumptions, they were transformed by logarithm for analysis, but the data are presented in their non-transformed values for ease of interpretation.

## 3. Results

### 3.1. Analysis of the Cytotoxicity of the Plant Extract Solutions in HEp-2 Cells

Initial experiments were performed to evaluate the cytotoxic potential of the plant extracts on HEp-2 cells in vitro (Figure 1). When treated with medium containing up to a quarter (0.25) plant extract solution, spicebush (*Lindera benzoin* (L.) *Blume*) and sage (*Salvia lyrata* (L.)) extracts showed a dose-dependent decline in cell viability with slopes that were significantly different compared to water control (ANCOVA, *p* < 0.0001). In contrast, the cell viabilities relative to untreated controls for cells treated with either aster (*Symphyotrichum novae-angliae* (L.) *Nemsom*) or lamb’s ear (*Stachys byzantina* (K.) *Koch*) extracts remained at or above 100% at all tested concentrations and showed no significant difference in slope compared to water treatment. For spicebush and sage extract treatments, the extract concentration associated with a 50% reduction in cell cytotoxicity (CC_50_) for HEp-2 cells was calculated to be 10.12 mg/mL of sage extract and 26.10 mg/mL of spicebush extract.

### 3.2. Impacts of Plant Extract Solutions on the Morphology and Proliferation of HEp-2 Cells

To determine the impact of the plant extracts on cancer cell morphology in vitro, immunofluorescence was performed on HEp-2 cells treated with medium supplemented with 10% of the plant extract solutions (Figure 2). Compared to untreated control, a reduction in the amount of visible cells and alterations in cell morphology (e.g., rounding and shrinking) consistent with cell death were observed when treated with colchicine, a known inducer of apoptosis and positive control (Figure 2A). Consistent with the lack of cytotoxicity previously observed, there were no major differences in the amount of growth observed when comparing either aster or lamb’s ear treatments to the untreated control. In contrast, cell volume and global cell health appeared diminished during treatment with either sage or spicebush extracts. These cells appeared to more closely resemble the colchicine treatment.

Using blinded scoring of nuclear morphology from DAPI-stained images, the number of cells exhibiting altered nuclear morphology (consistent with activation of cell death pathways) and cells undergoing mitosis was determined for each of the treatments (Figure 2B). Compared to untreated controls, there were significantly greater amounts of cells with altered nuclear morphology in the colchicine, sage, and spicebush conditions (ANOVA, *p* < 0.01 for all conditions). The average percentages of cells with altered nuclear morphology during treatments with sage (12.2%) and spicebush (14.1%) were 4.1× and 4.7× the rates observed in untreated cells. Furthermore, comparisons between the amount during either treatment and colchicine (14.3%) revealed no significant difference. In contrast, no significant differences were observed between aster (1.7%), lamb’s ear (2.4%), and the untreated control (3.0%). Analysis of the mitotic index revealed significant reductions in mitosis observed when comparing colchicine (0.6%, *p* = 0.018) and spicebush (0.7%, *p* = 0.022) to the untreated control (4.1%). No other treatments resulted in significant reductions in mitosis. A similar blinded analysis was performed while examining the actin and tubulin cytoskeletal morphology of treated HEp-2 cells (Figure 2C). Cells treated with colchicine, sage, and spicebush exhibited significantly greater cytoskeletal disruption consistent with progression toward apoptosis (ANOVA, *p* < 0.001 for all 3 treatments).

While these observations are consistent with induction of apoptosis, it is possible that other cell death pathways (e.g., necroptosis, ferroptosis) were being activated. An activation assay for caspases 3 and 7 was performed on cells treated with the plant extracts for 4 h (Figure 2D). Compared to untreated controls, sage (164%, *p* = 0.00186) and spicebush (176%, *p* = 0.0098) extracts elicited significantly greater caspase 3 and 7 activity, whereas no significant change was detected in cells treated with either aster (93%) or lamb’s ear (84%). Collectively, these data are consistent with sage and spicebush treatments promoting activation of apoptosis.

### 3.3. Antiviral Activity of the Plant Extracts against Human Respiratory Syncytial Virus (hRSV)

hRSV remains a major pathogen of children and the elderly and is a common circulating respiratory pathogen. Several of the plant extracts being tested were traditionally used for respiratory ailments. Experiments were performed to evaluate whether the plant extracts were capable of inhibiting the infectivity of hRSV (Figure 3). To avoid the impact of cytotoxicity on HEp-2 cell viability, cells were treated with proportions of plant extract solutions at or below 2.5% during infection with hRSV at an MOI of 0.05. Significant dose-dependent reductions in hRSV infectivity were observed with aster, sage, and spicebush treatments relative to a water-treated control (ANCOVA, *p* < 0.0001). No significant reduction was observed between lamb’s ear and the water-treated control. The effective concentration to reduce viral infectivity by 50% (EC50) was calculated for each of the treatments. Consistent with the significant inhibition observed, the EC50 values for aster and sage were found to be 0.16 mg/mL and 0.99 mg/mL, respectively. While significant compared to control, the concentration of spicebush needed to reduce viral infectivity by 50% was found to be greater at 3.60 mg/mL.

### 3.4. Antiproliferative and Antiviral Activity of Major Compounds Identified in the Plant Extracts

Each of the plant extracts were analyzed by GC-MS after extraction in ethanol (Figure 4). Three different prominent compounds were identified in sage extract: (1S)-(−)-borneol, 1,8-cineole (eucalyptol), and D-(+)-melezitose. In spicebush extract, 5-(hydroxymethyl)-2-furaldehyde was identified as a major component. Lastly, D-(+)-melezitose was also identified in aster extract. No prominent compounds were identified in lamb’s ear extract.

Each of the major compounds were analyzed for cytotoxicity at concentrations ranging from 0.001 times to 10 times the amount present in the plant extracts from which they were derived (Figure 5). All four compounds induced reductions in cell viability at the 1× concentration (equivalent to the relative concentration present in the plant extract) with cell viability. However, the cell viability remained greater than 61% for all compound treatments at 1× concentration. Consistent with this observation, the CC50 was determined for each of the treatments, and all treatments required greater than 1× concentrations, which ranged from 5.74× (furfural) to 19.09× (melezitose), to induce a 50% reduction in cell viability. When 10× concentrations were used, the cell viability was reduced by approximately 80% for borneol (23% viability at 77 mM), cineole (22% at 52 mM), and furfural (23% at 48 mM) treatments, and approximately 25% for melezitose treatment (73% viability at 10 mM). Collectively, none of the pure compounds exhibited comparable cytotoxicity to the extracts from which they were each derived (Figure 2).

Immunofluorescence analysis was performed on each of the compounds to evaluate any impacts that they have on cell morphology or induction of apoptosis (Figure 6). No significant differences were observed in the immunofluorescence images obtained for DNA, actin, or tubulin staining (Figure 6A). Consistent with the images, blind scoring of the number of cells with altered nuclei or mitosis revealed no significant differences. The average percent of cells with altered nuclei ranged from 1.2% (melezitose) to 2.1% (borneol) for compound treatments compared to the control treatment (1.1%) (Figure 6B). The average percent of cells in mitosis ranged from 4.1% (borneol) to 6.6% (furfural) for compound treatments compared to the control treatment (7.4%). Lastly, none of the treatments resulted in an average cytoskeletal alteration score greater than 2 indicating no appreciable disruption of cytoskeleton arrangement among the treatments (Figure 6C).

Each of the major compounds from the plant extracts was also evaluated for antiviral activity against hRSV at concentrations ranging from 0× to 2× levels found in the plant extracts (Figure 7). At the 1× concentration, which is equivalent to the amount of compound present in the extracts tested, treatment with only two compounds resulted in significant reductions in virus: borneol and furfural (ANCOVA, *p* < 0.0001). The EC50 for each of the major compounds was calculated, and 50% reductions in virus were observed at 0.12× (0.57 mM) and 0.45× (3.47 mM) amounts of furfural and borneol, respectively, relative to their actual concentrations in the extract. In contrast, concentrations greater than amounts observed in the extract, were required to induce a 50% reduction in virus for cineole (1.32× or 6.73 mM) and melezitose (2.54× or 2.54 mM).

## 4. Discussion

The plants used in the current study have been used medicinally for many years. While previous studies have addressed cytotoxic or antiviral properties of aqueous extracts of some of these plants, the purpose of this study is to compare the effects of the individual plants to each other as well as begin the elucidation of the primary active components within the extracts [24,25,26,27,28,29,30]. This analysis may be a direct step toward the future of antiviral and anti-cancer therapeutics, while also leading the way in further research into RSV vaccinations.

An in-depth exploration of cytotoxicity revealed distinct impacts of various extracts on HEp-2 cells. Notably, sage and spicebush extracts exhibited pronounced effects on cell viability compared to aster, lamb’s ear, and the control (Figure 1). Nuclear staining indicates that sage and spicebush extracts cause this decrease in cell viability both by increasing apoptosis and decreasing mitosis. This was confirmed by scoring of the cytoskeletal disorganization (Figure 2). Both mitosis and apoptosis are processes governed by morphological changes to the cytoskeleton with cytoskeletal organization in the former and disorganization in the latter [31,32,33]. These findings suggest sage and spicebush could potentially be used as a cancer therapeutic and are similar to previous findings [27,29].

HEp-2 cells were also infected with RSV and treated with plant extracts to investigate the impact each extract had on viral inhibition. The plant extracts were used at concentrations below those found to significantly reduce cell numbers to ensure that the change in viral count was due to viral inhibition and not HEp-2 cell cytotoxicity. Aster and sage displayed concentration-dependent viral inhibition which has been reported previously (Figure 3) [25,26]. These findings suggest the potential of aster and sage extracts in curbing RSV infection, with early inhibitory effects observed even at lower concentrations. The data also indicate that because aster extract showed no evident cytotoxicity but did have antiviral activity, it may have a better safety profile as a possible therapeutic.

A pivotal step in exploration involved investigating the individual contributions of major compounds from each plant extract on HEp-2 HeLa cell cytotoxicity. When identifying major compounds, a relative amount of compound in the extract was also calculated. Cells were then treated with a range of compound concentration from 0× to 2× the levels found in the plant extract. Surprisingly, our findings indicated that the isolated major compounds did not significantly influence cell viability (Figure 5) or cytotoxic activity (Figure 6). This implies that the observed antiviral and anti-cancer effects stem from a collective action of compounds rather than any singular leading compound. Alternatively, it remains possible that outcomes observed from the whole extracts may be due to compounds yet to be isolated.

When exploring the antiviral activity of major compounds, borneol and furfural were both able to significantly inhibit viral replication (Figure 7). Borneol was originally isolated from the sage extract, and furfural was isolated from the spicebush extract. Borneol’s ability to inhibit viral replication could be the source of sage’s antiviral activities and has been previously shown to inhibit RSV fusion [34]. It should be noted that borneol is a common constituent in many essential oils and has a wide range of therapeutic potential, including antiviral activities [34,35,36]. It is also possible that there are other compounds that are yet to be isolated that have an additive effect with the borneol. Furfural is interesting, however, since the complete spicebush extract did not show significant antiviral activity even though furfural was able to reduce viral replication at below 1x concentrations. This could be because the antiviral effects of furfural are reduced with the combination of other compounds due to inactivation or blocking of a specific binding site. Melezitose, which was isolated from aster, and cineole, which was isolated from sage, did not show any significant antiviral activity. This indicates that there are other compounds within those extracts that lead to the antiviral properties. The lack of significant activity associated with cineole (commonly known as eucalyptol) is particularly surprising due to its common use and broad therapeutic activities, which include pro-apoptotic effects and use in treating respiratory ailments [37]. Our findings may indicate that the concentration of cineole identified in our plant extracts was insufficient to elicit these biologic activities. While the major compounds described here were largely unique to each plant extract, it would be interesting in future studies to evaluate any therapeutic potential associated with combining these major constituents together for potential synergistic activities.

## 5. Conclusions

In conclusion, our investigation highlights aster’s antiviral activity, spicebush’s anti-cancer potential, and sage’s dual attributes. These findings offer promising leads for treating RSV and cancer, holding the potential to significantly enhance global human well-being. Despite our effort to isolate and assess major compounds, the precise molecules driving the observed antiviral and anti-cancer effects remain an intriguing question. Future studies may unravel these mysteries, leading to the development of innovative therapies and vaccination strategies.

This research contributes to the growing understanding of formulated plant extracts’ potential in addressing pressing medical challenges. The intricate interplay of compounds and their multifaceted activities underscore the complexity of natural therapeutics, urging further interdisciplinary collaboration and exploration. As we embark on this scientific journey, the quest for unlocking the precise mechanisms underlying these observed effects remains an exciting avenue for future research.

## Figures and Tables

**Figure 1 biology-13-00696-f001:**
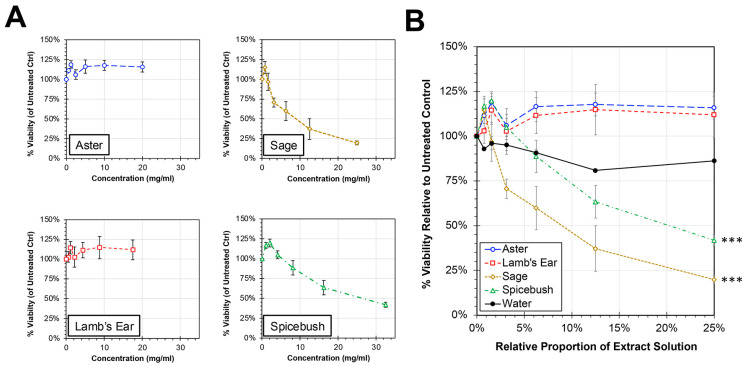
**Cytotoxicity of Plant Extracts in HEp-2 cells**. HEp-2 cells were treated with cell culture medium containing proportions of plant extracts between 0 and 25% for 24 h. After 24 h, the cell viability was determined by MTS assay, and average relative viability (±std error of the mean; N = 4) compared to untreated controls is shown. The individual extracts shown in units of concentration (**A**) and by proportion of treatment solution (**B**). An ANCOVA was performed to compare the slopes of each treatment to the water control and significance is indicated (***, *p* < 0.0001).

**Figure 2 biology-13-00696-f002:**
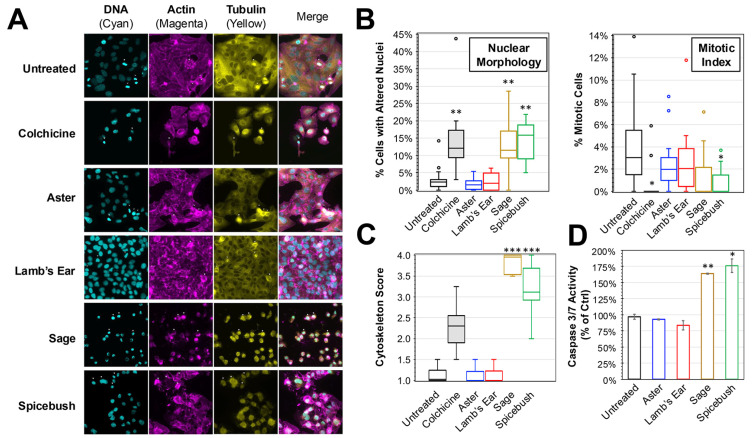
**Immunofluorescent Analysis of HEp-2 Cells Treated with Plant Extracts.** (**A**) Immunofluorescence images were taken of HEp-2 cells treated after 24 h incubation in medium containing 10% added extract solutions. DAPI (staining DNA), TRITC-phalloidin (actin), and FITC-conjugated anti-tubulin (tubulin) were used and the separate and merged images are shown. (**B**) The number of cells with altered nuclei (left) and undergoing mitosis (right) was determined by blinded scoring of DAPI-stained nuclei images (N > 12) of treated cells. (**C**) A cytoskeleton score for the amount of cytoskeletal disruption in each image (N > 12) based on blinded analysis of the actin and tubulin morphology is shown. The box and whisker plots depict the boxes with lines for the median, 25th, and 75th percentiles, as well as whiskers for the 5th and 95th percentiles. Outliers are shown as individual points. One-way ANOVA was performed to compare the nuclei, mitotic index, and cytoskeleton scores to the untreated control. (**D**) Activation of caspase-3/-7 after 4 h of treatment with 10% extract solutions as determined. Average percent (±SD) of untreated control activity is shown. The significance is noted for all experiments (*, *p* < 0.05; **, *p* < 0.01; ***, *p* < 0.001).

**Figure 3 biology-13-00696-f003:**
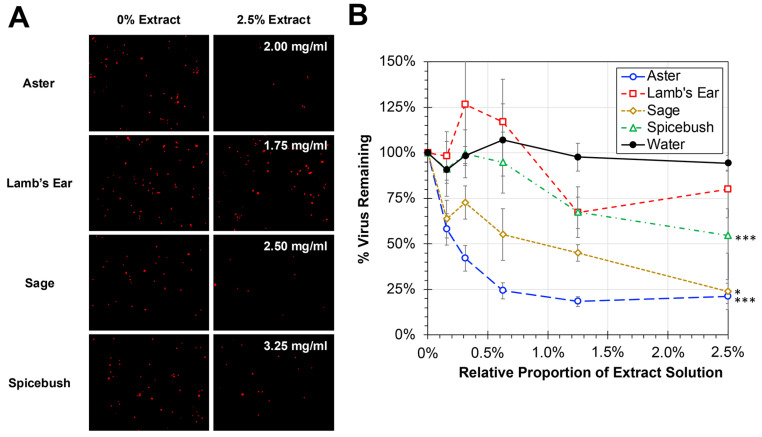
**Antiviral Activity of Plant Extracts Against hRSV.** (**A**) HEp-2 cells were infected at an MOI of 0.05 with hRSV strain A2-mKate2 in cell culture medium containing proportions of plant extracts between 0 and 2.5% for 24 h before images were obtained and the amount of hRSV-infected cells quantified and normalized to no extract treatment. The concentration of plant extract for 2.5% used is shown as an inset within the treatment images. (**B**) The average percent of remaining virus relative to no treatment is shown (±std error of the mean; N = 3). An ANCOVA was performed to compare the slopes of each treatment to the water control and significant reductions are indicated (*, *p* < 0.05; ***, *p* < 0.001).

**Figure 4 biology-13-00696-f004:**
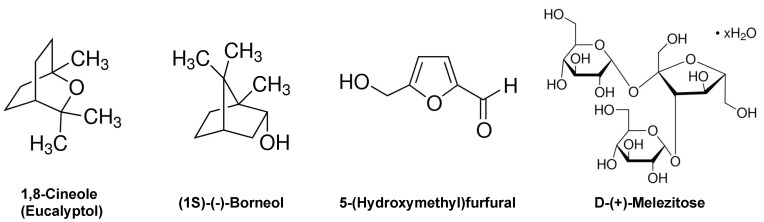
**Major Compounds in Plant Extracts Identified by GC-MS Analysis**. Chemical structures of the major compounds identified by GC-MS analysis are shown.

**Figure 5 biology-13-00696-f005:**
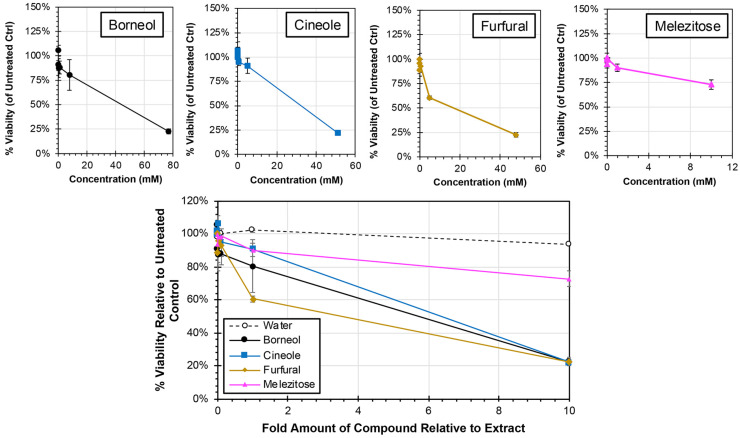
**Cytotoxicity of Major Compounds from Extracts in HEp-2 cells**. HEp-2 cells were treated with cell culture medium containing major compounds or a water control at concentrations ranging from 0.001× and 10× (individual concentration curves are shown above) of their concentrations in the plant extracts. Individual cytotoxicity curves as a factor of concentration are shown above. After 24 h, the cell viability was determined by MTS assay, and average relative viability (±std error of the mean; N = 3) compared to untreated controls is shown.

**Figure 6 biology-13-00696-f006:**
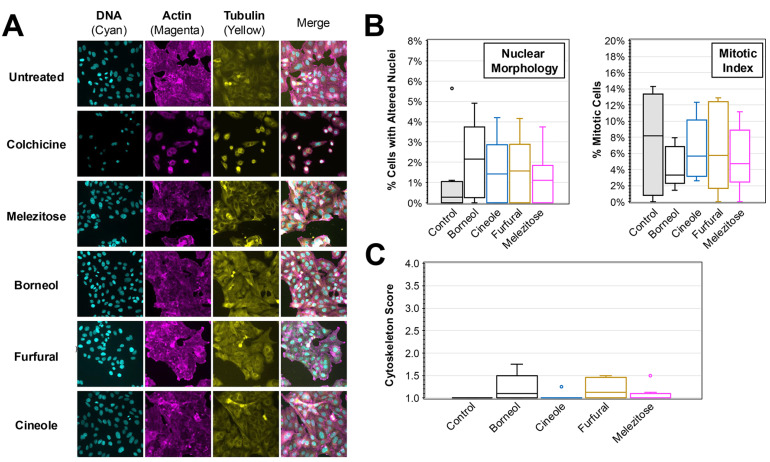
**Immunofluorescent Analysis of HEp-2 Cells Treated with Major Compounds from Plant Extracts**. (**A**) Immunofluorescence images were taken of HEp-2 cells treated after 24 h incubation in medium containing major compounds at the concentration present in plant extracts. DAPI (staining DNA), TRITC-phalloidin (actin), and FITC-conjugated anti-tubulin (tubulin) were used, and the separate and merged images are shown. (**B**) The number of cells with altered nuclei (left) and undergoing mitosis (right) was determined by blinded scoring of DAPI-stained nuclei images (N > 8) of treated cells. (**C**) A cytoskeleton score for the amount of cytoskeletal disruption in each image (N > 8) based on blinded analysis of the actin and tubulin morphology is shown. The box and whisker plots depict the boxes with lines for the median, 25th and 75th percentiles, as well as whiskers for the 5th and 95th percentiles. Outliers are shown as individual points. One-way ANOVA was performed to compare the nuclei, mitotic index, and cytoskeleton scores to the untreated control, and no significance is noted.

**Figure 7 biology-13-00696-f007:**
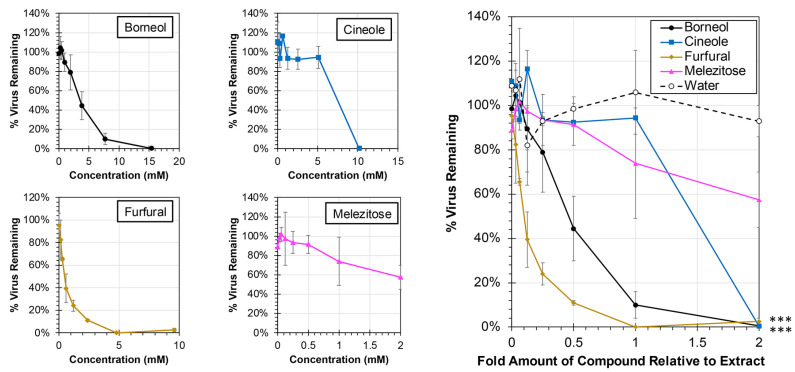
**Antiviral Activity of Major Compounds from Plant Extracts Against hRSV**. HEp-2 cells were infected at an MOI of 0.05 with hRSV strain A2-mKate2 in cell culture medium containing proportions of the major compounds between 0× and 2× relative to extract concentration (individual concentration plots are shown at left) for 24 h before images were obtained and the number of hRSV-infected cells quantified and normalized to no extract treatment. The average percent of remaining virus relative to no treatment is shown (±std error of the mean; N = 2). An ANCOVA was performed to compare the slopes of each treatment to the water control and significance is indicated (***, *p* < 0.0001).

## Data Availability

All data and procedures described herein are available upon request.

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
