# Peer review of "Evaluation of the Therapeutic Potential of Traditionally-Used Natural Plant Extracts to Inhibit Proliferation of a HeLa Cell Cancer Line and Replication of Human Respiratory Syncytial Virus (hRSV)"

_biology, 2024, doi:10.3390/biology13090696_

Round 1
Reviewer 1 Report
Comments and Suggestions for Authors
I am very sorry to reject this work, but in my opinion, several improvements are needed.
Majors
- No experiments (western blots, immunofluorescence, etc.) have been carried out to investigate, at least in part, the mechanism of the extracts (both antiviral and antitumour). You only reported some alterations in apoptosis, mitosis and the cytoskeleton (but even here I have some concerns which I report in the next point).
- I assess mitotic cells using bromodeoxyuridine, propidium iodide, clonogenic assay, CFSE, mitotic markers, etc., but the use of dapi alone seems to me too reductive, to determine both mitotic and apoptotic cells, among other things. Dapi increases inter alia its fluorescence during apoptosis: for example, in Figure 2, the image of the lamb's ear appears to be more fluorescent than the control, but the levels of apoptosis are the same. For this reason, fluorescence microscopy is not particularly suitable for determining these parameters. Confocal microscopy is necessary to eliminate artefacts caused by out-of-focus light, allowing much sharper and more detailed images of the focal plane (ctrl cells are in the other z-plane appearing less bright, but are alive like these very bright in Lam's ear). This is also particularly relevant to appreciate precisely the rearrangements of the cytoskeleton. On the other hand, I could have accepted these data (using confocal microscopy) if supported by another experiment at least (evidence of apoptotic markers activation as caspase cleavage, PARP, Bcl2 family, evidence of rearrangement of stress fibers, etc.). By the way, you cannot talk about apoptosis using dapi alone, it could also be necroptosis, ferroptosis or other forms of cell death leading to irregular and fragmented nuclei. So I would talk about "cell death" or "hypodiploid nuclei" with this information alone.
- A complete characterization of the extracts is necessary. Borneol and eucalyptol are very common in many essential oils.
Minors
- I do not understand why no concentrations are reported but only relative proportions. I find it difficult to understand what concentrations are tested without seeing them in the graphs.
- The introduction is too weak on the part that connects tumour and this virus compared to others. I would better motivate why you wanted to evaluate both
- In order to assess whether the majority substances together might work/not work, I would also test their mixes and not only individually
- Indicate the concentrations used of the extracts in materials and methods, and in the results
Author Response
Majors
- No experiments (western blots, immunofluorescence, etc.) have been carried out to investigate, at least in part, the mechanism of the extracts (both antiviral and antitumour). You only reported some alterations in apoptosis, mitosis and the cytoskeleton (but even here I have some concerns which I report in the next point).
While we fully appreciate and agree with your point here to have pursued the exact mechanisms underlying the nature of these extracts, given the original size and scope of this project, we were aiming to evaluate if there were any notable changes on cancer cell viability or virus susceptible in response to each of the plant extracts studied. It is our hope in the future to assess the molecular mechanisms underlying the significant outcomes identified here. In response to your specific concern regarding whether the plant extracts were inducing apoptosis, we were able to perform a Caspase-3/-7 assay and found that both sage and spicebush elicited significantly higher activation relative to untreated controls (lines 259 - 266; Figure 2).
- I assess mitotic cells using bromodeoxyuridine, propidium iodide, clonogenic assay, CFSE, mitotic markers, etc., but the use of dapi alone seems to me too reductive, to determine both mitotic and apoptotic cells, among other things. Dapi increases inter alia its fluorescence during apoptosis: for example, in Figure 2, the image of the lamb's ear appears to be more fluorescent than the control, but the levels of apoptosis are the same. For this reason, fluorescence microscopy is not particularly suitable for determining these parameters. Confocal microscopy is necessary to eliminate artefacts caused by out-of-focus light, allowing much sharper and more detailed images of the focal plane (ctrl cells are in the other z-plane appearing less bright, but are alive like these very bright in Lam's ear). This is also particularly relevant to appreciate precisely the rearrangements of the cytoskeleton. On the other hand, I could have accepted these data (using confocal microscopy) if supported by another experiment at least (evidence of apoptotic markers activation as caspase cleavage, PARP, Bcl2 family, evidence of rearrangement of stress fibers, etc.). By the way, you cannot talk about apoptosis using dapi alone, it could also be necroptosis, ferroptosis or other forms of cell death leading to irregular and fragmented nuclei. So I would talk about "cell death" or "hypodiploid nuclei" with this information alone.
Thank you for addressing this point. While we were not able to obtain confocal images of the treatments, we have revised the language describing the interpretation of the nuclei morphology and did perform a Caspase-3/-7 assay to address whether the alterations in cell morphology were due to induction of apoptosis (in contrast to the other mechanisms (e.g. necroptosis and ferroptosis) mentioned here (lines 241 – 266 and lines 355 - 361; Figures 2 and 6).
- A complete characterization of the extracts is necessary. Borneol and eucalyptol are very common in many essential oils.
We have elaborated on these substances in the discussion section and have included additional references related to their known and previously described biological activities (lines 449 - 463).
Minors
- I do not understand why no concentrations are reported but only relative proportions. I find it difficult to understand what concentrations are tested without seeing them in the graphs.
We have included concentrations in parallel to relative proportions throughout the manuscript and have amended all graphs to include concentrations of both plant extracts and isolated major compounds (lines throughout; Figures 1 – 3 and 5 – 7).
- The introduction is too weak on the part that connects tumour and this virus compared to others. I would better motivate why you wanted to evaluate both
We have added several sentences to the end of the introduction which establish the rationale for use of both our HeLa cell line and hRSV as test subjects for this study (lines 90 - 95).
- In order to assess whether the majority substances together might work/not work, I would also test their mixes and not only individually.
Since the compounds were identified in separate plants, we wanted to identify whether each major compound was responsible for the activities shown. This would be a great future direction and we have made reference to it in the discussion. (lines 464 - 467).
- Indicate the concentrations used of the extracts in materials and methods, and in the results
We have added the concentrations of both extracts and identified major constituent compounds into the materials and methods and as previously mentioned, have also included concentrations throughout the results (lines throughout with specific changes in methods here of lines 138 - 141; Figures 1 – 3 and 5 – 7).
Reviewer 2 Report
Comments and Suggestions for Authors
I only detect minimal grammatical errors in some expressions, such as in line 25 to cite an example. “remains Limited Scientific”… I find it more convenient to mention “Remains a Limited”… or mention “the Limited ..."
Author Response
I only detect minimal grammatical errors in some expressions, such as in line 25 to cite an example. “remains Limited Scientific”… I find it more convenient to mention “Remains a Limited”… or mention “the Limited ..."
Thank you for your recommended edit and feedback. We have made this change and have corrected a few additional small grammatical errors throughout the manuscript on a second read (lines 25 - 26).
Reviewer 3 Report
Comments and Suggestions for Authors
Interesting paper, very well written, but with a few aspects that must be improved/corrected:
- Title: since only one cell line was used I strongly suggest a title adjustment, so it reflects the study in a more accurate way;
- abstract: it is suggested that the abstract is rewritten, in order to include more results and a conclusion (suggestion: shorten the introduction)
- introduction: line 44 - remove the words "to date" oh the phrase, since the article cited is from 2016
lines 84-86 - the sentence must be rewritten or removed, since it is not an introduction to the theme, but, instead, more of a conclusion
- material and methods: it is supposed that the plants have a voucher ID attributed (since they were identified by an ethnobotanist (line 103); that information must be, however, clear in the article
there is lacking information regarding the plant extraction method, namely time of extraction (line 109)
- results: the expression "in vitro" must always be in italic; verify and correct throughout the article (e.g. line 204)
the use of subjective adjectives should be avoided; when they exist, the language must be corrected ( e.g. line 206 - dramatic reduction)
when the authors refer to "overall cell health" (line 207) they must be more specific/objective
Comments on the Quality of English Languageminor editing in the use of a more objective language must be undertaken
Author Response
- Title: since only one cell line was used I strongly suggest a title adjustment, so it reflects the study in a more accurate way;
We appreciate the reviewer’s suggestion and have changed the title to reflect the specific cell line tested with this study (Title).
- abstract: it is suggested that the abstract is rewritten, in order to include more results and a conclusion (suggestion: shorten the introduction)
Thank you for your recommendation. We have shortened the introduction and have rewritten the second half of the abstract to include more results and findings from our study (lines 24 - 41; Abstract)
- introduction: line 44 - remove the words "to date" oh the phrase, since the article cited is from 2016
We have made this recommended change ( lines 49 - 50).
-lines 84-86 - the sentence must be rewritten or removed, since it is not an introduction to the theme, but, instead, more of a conclusion
We have removed this sentence from the introduction.
- material and methods: it is supposed that the plants have a voucher ID attributed (since they were identified by an ethnobotanist (line 103); that information must be, however, clear in the article
We have included the ID numbers for the plant specimens that are archived in the Freisner Herbarium and are available upon request (lines 115 - 118).
-there is lacking information regarding the plant extraction method, namely time of extraction (line 109)
Thank you for pointing this out. We have now included a statement concerning the timing of the extraction (lines 122 - 123).
- results: the expression "in vitro" must always be in italic; verify and correct throughout the article (e.g. line 204)
We have italicized all instances of in vitro throughout the manuscript.
-the use of subjective adjectives should be avoided; when they exist, the language must be corrected ( e.g. line 206 - dramatic reduction)
We have rewritten this part of the sentence to be more clear and less subjective (lines 231 - 234).
-when the authors refer to "overall cell health" (line 207) they must be more specific/objective
We have rewritten this part of the sentence to be more clear and less subjective (lines 231 - 234).
Round 2
Reviewer 1 Report
Comments and Suggestions for Authors
The manuscript was improved, it can ben accepted in the current form.